# Analysis of Phenolic Composition of Byproducts (Seeds and Peels) of Avocado (*Persea americana* Mill.) Cultivated in Colombia

**DOI:** 10.3390/molecules24173209

**Published:** 2019-09-03

**Authors:** Johanna C. Rosero, Silvia Cruz, Coralia Osorio, Nelson Hurtado

**Affiliations:** 1Departamento de Química, Universidad de Nariño, Pasto AA 1175, Nariño, Colombia; 2Departamento de Química, Universidad Nacional de Colombia, Bogotá DC AA 14490, Colombia

**Keywords:** polyphenols, Trolox equivalent antioxidant capacity (TEAC), DPPH, UPLC-ESI-MS/MS, bio-guided fractionation

## Abstract

The byproducts (seeds and peels) of an avocado cultivated in the south of Colombia were extracted with aqueous acetone and their antioxidant properties were measured with ABTS (2,2′-azino-bis(3-ethylbenzothiazoline-6-sulphonic acid) and DPPH (2,2-diphenyl-1-picrylhydrazyl) assays, and total polyphenol content was determined by Folin–Ciocalteu method. A bioguided fractionation was performed, first by SPE (solid phase extraction) on Amberlite XAD-7, and then by size exclusion chromatography on Sephadex LH-20. The polyphenolic-rich extracts and their fractions were analyzed by ultra-performance liquid chromatography–electrospray ionization–mass spectrometry (UPLC-ESI-MS/MS), finding the presence of organic acids, hydroxycinnamic acids, catechins, free and glycosylated flavonoids, and dimeric and trimeric procyanidins. Catechin, epicatechin, six quercetin derivatives, four dimeric procyanidins (three type B and one type A), and three trimeric procyanidins (two type B and one type A) were detected in the most active fractions of avocado peel and seeds. The most antioxidant fractions contain the higher molecular weight phenolic compounds (condensed tannins).

## 1. Introduction

Avocado is a tropical fruit native to America that is commonly distributed from the northern region of South America to Mexico [1]. It is a species that belongs to the Laureaceae family and the *Persea* genus, which includes three species, *P. schiedeana*, *P. parvifolia,* and *P. americana*. The last one (avocado) exhibits a series of phenotypic variants due to different geographical, climatic, genetic and evolutionary conditions [2]. This species has eight subspecies, which climatic requirements and zones of origin of which have been well-defined; among them, five have no any commercial value, while the other three variants, *drymifolia, guatemalensis, and americana*, correspond to the widely cultivated avocado and are commonly known as Mexican, Guatemalan and Antillean breeds, respectively. These three breeds are able to hybridize easily, obtaining trees with a higher adaptability to different climates and agronomic characteristics. The Mexican and Guatemalan genotypes, along with their hybrids, are adaptable to subtropical zones, while the Antillean only grows in tropical areas [3].

The avocado fruit is a drupe constituted by epicarp (peel), mesocarp (pulp), and endocarp (seed), whose size, shape, color and phytochemical content depend on the genotype. The peel color can be light green, dark green, purple or black and it can present a smooth, rough, lustrous or opaque appearance. Also, the fruit weight may fluctuate between 100 and 3000 g [4]. Recently, the physicochemical characterization of 21 commercial varieties of avocado from Colombia (subspecies *guatemalensis*, *americana* and their hybrids) were performed, evaluating the fruit morphological, features, nutritional value (i.e., monounsaturated fatty acids, phytosterols, and vitamin E amounts), and metabolic profile of the mesocarp oil. It was found that similar physicochemical characteristics are mainly influenced by genetic factors, and by environmental and geographic conditions to a lesser extent [5]. According to Ministry of Agriculture, Colombia is nowadays a world leader in the production of avocado (var. Hass) with 400,000 ton/year, of which peels constitute between 11.9–16.1% *w*/*w* and seeds between 16.1–20.9% *w*/*w*.

Previous studies have shown that the avocado fruit is a rich source of polyphenolic compounds that are considered human health-promoting [6]. In fact, there is evidence showing that these polyphenols exhibit antibacterial, anti-allergenic, platelet aggregation inhibitor, antihypoglycemic [7], anti-oxidant, anti-inflammatory [8], and chemoprotective against cancer [9] activities, and also improve lipid metabolism. Also, avocado seeds and peels have been reported as a good source for cosmetic products, including the behenyl alcohol in the oil. For this reason, there is an increasing interest to use these compounds in food, nutraceutical and/or pharmaceutical industries. In addition to their potential health benefits, antioxidant compounds could be used to delay the lipid oxidative damage that leads food rancidity. The compounds associated with these biological effects are mainly phenolic acids and flavonol derivatives [10,11]. Phenolic compounds can be found as conjugated (soluble) or linked forms (insoluble). The polymeric forms of phenolic compounds are associated with the cell wall and their release require acid, basic or enzymatic hydrolysis, which differ from the extraction of free polyphenols that can be done by using polar solvents such as, methanol, ethanol or acetone.

Byproducts from avocado (seeds and peels) are a good source of phenolic compounds, and their effectiveness as inhibitors of lipid and protein oxidation and color deterioration in raw porcine patties has been studied. The extracts of “Hass” and “Fuerte” avocado varieties showed to be natural antioxidants that enhance the quality of muscle foods [6,12]. Avocado peels and seeds showed higher amounts of phenolics than the pulp. Avocado pulp is rich in hydroxybenzoic and hydroxycinnamic acids and procyanidins; while in seeds, the main antiradical capacity was attributed to the presence of (+)-catechin, (−)-epicatechin, and 3-*O*-caffeoylquinic acid, among other flavonoid compounds, such as procyanidins and hydroxycinnamic acids [13]. Recently, avocado peel (var. Colinred) methanolic extract showed an in vivo antioxidant capacity (by using *Drosophila melanogaster*), related to the prevention of Parkinson’s disease, and against induced oxidative stress [14].

Even though previous studies on avocado byproducts (seeds and peels) have contributed significant information regarding their antioxidant properties, the identity of the individual phenolic compounds responsible for that activity still remains unclear. Additionally, most of the avocado byproduct extracts have been obtained by using either maceration [15,16] or accelerated solvent extraction [6,17]. Thus, the aim of this work was to perform a bioguided fractionation of polar extracts from peels and seeds of avocado by using a combination of different chromatographic techniques (selective solid phase extraction (SPE) and size exclusion chromatography), and identify some of the phenolic compounds responsible for the antioxidant capacity exhibited by these byproducts.

## 2. Results and Discussion

### 2.1. Antioxidant Capacity of Avocado Extracts and Fractions

There are around 400 varieties of avocado, resulting from an open polinization and hybridization between subspecies, making their phylogeny a difficult topic to study. In the present study, the polyphenolic profile of peels and seeds of avocado cultivated in Southwestern Colombia were analyzed. This cultivar is characterized by a bright green and rugose peel that covers the yellow-green pulp rich in oil. The large seed and peel account for an average of 15.5% and 9.3% of fruit’s weight, respectively (Figure 1).

For this purpose, two types of extraction solvents (aqueous methanol 80% and aqueous acetone 70%) were used; based on the results of antioxidant capacity, aqueous acetone was chosen for further experiments. However, it was found that the fruit part significantly affects the value of antioxidant capacity (*p* < 0.05, 95% confidence). Thus, samples were initially subjected to chemical maceration in 70% acetone, the solvent was removed under vaccum, and the final extracts—crude seed extract (CSE) and crude peel extract (CPE)—were obtained by lyophilization. The crude extracts were separately passed through a column packed with XAD-7 Amberlite to get the phenolic-rich seed extract (PSE, 52.5% CSE), and the phenolic-rich peel extract (PPE, 65.5% CPE), respectively. 

To obtain less-complex fractions, the PSE and PPE extracts were fractionated by using a Sephadex LH-20 column, with aqueous ethanol and aqueous acetone as eluents. This method was effective to isolate low, medium, and high molecular mass polyphenols. It has been found that the polyphenols containing proanthocyanidins are partly separated according to the difference of affinity force to the gel matrices [18]. Finally, three fractions from each extract were collected: F1S (12%), F2S (19.9%), and F3S (62.7%) fractions from PSE, and F1P (6.4%), F2P (21.5%), and F3P (62.8%) from PPE.

The total phenolic content (TPC) of extracts and fractions was assessed by using the Folin–Ciocateau method, and the antioxidant capacity was determined by the ability to scavenge the ABTS (2,2′-azino-bis(3-ethylbenzothiazoline-6-sulphonic acid) and DPPH (2,2-diphenyl-1-picrylhydrazyl) radical cations (Table 1). Calculation of Fisher significant minimum difference procedure (LSD) showed a statistically significant difference between samples in the total phenolic contents (*p* < 0.05). It was also statistically established that the phenolic-rich extracts (PSE and PPE) exhibited a higher antioxidant capacity value than the corresponding crude extracts (CSE and CPE), thus showing the effectiveness of fractionation process.

The results of antioxidant capacity showed a similar behavior than TPC. For example, TEAC values of PSE and PPE were 5.8- and 2.1-fold higher than those determined for the respectively CSE and CPE, being PSE the extract with the highest antioxidant capacity. On the other hand, a statistically significant relationship (*p* < 0.05, 95% confidence) between the TPC values and the antioxidant capacity values (TEAC) from the phenolic-rich extracts (PSE and PPE) (*r* = 0.9695 and 0.9846, respectively) was found. Nonetheless, no relationship was found between the TPC data and the DPPH antioxidant value (IC_50_) of PPE (low correlation coefficient, *r* = 0.6500), attributable to the existence of a competitive reaction kinetics between antioxidants and the substrate. The in vitro antioxidant capacity of avocado seed extract has been before reported, showing a higher potential than the fruit pulp, due to its tannin and polyphenol contents [6,19]. Sang Vo et al. [20] reported not only the free radical scavenging capacity of avocado seed ethanolic extract (from Vietnam) and their fractions, but also their inhibitory effect on proliferation of human lung A549 and human gastric BGC823 cancer cells.

The TPC and antioxidant capacity in peel and seeds of the different *Persea americana* cultivars have been reported in the literature. Slimcado, Hass, Tonnage, Loretta, and Simmonds seeds contained 19.2, 51.6, 33.1, 31.5, and 40.2 mg gallic acid equivalent (GAE)/g dry weight, respectively. On the other hand, for the same varieties the peel accounted for 4.6, 12.6, 4.3, 7.6, and 7.4 mg GAE/g dry weight, respectively [21]. Kosinska et al. [15] reported TPC values of 9.5 and 13.0 mg catechin equivalent/g dry weight for Hass and Shepard avocado seeds, respectively, and 25.3 and 15.6 mg catechin equivalent/g dry weight for peels. Thus, seeds of the Nariño cultivar contained more phenolic compounds with 328.8 mg GAE/g dried CSE (18.2 GAE/g dry seed) than seeds of Hass and Shepard; however, the TPC value for the peels, 527.8 mg GAE/g dried CPE, is smaller than those found for the Hass, Tonnage, Loretta and Simmonds cultivars. Various investigations have concluded that TPC values are highly affected by variety and agronomic conditions [21]. It was found that Nariño cultivar seeds had the highest ABTS radical scavenging capacity (3.2 mmol Trolox/g dried CSE (0.177 mmol Trolox/g dried seed)), in comparison to those exhibited by the extracts of Hass (0.094 mmol Trolox/g dry weight) and Shepard (0.091 mmol Trolox/g dry weight) seeds, and Shepard (0.112 mmol Trolox/g dry weight) peels [15]. Antasionasti et al. [16] studied the antioxidant capacity of avocado peels from Indonesia; they reported a significantly lower IC_50_ value (9.467 ±0.045 g/L) determined by DPPH assay for the methanolic extract in comparison with the results here reported (Table 1).

Additionally, the fractions F1S and F1P showed the lowest values of antioxidant capacity in comparison with the other fractions. Thus, it was evident that the most highly retained fractions (F2S, F2P, F3S and F3P) are the most bioactive under the tested synthetic radicals, in agreement with the TPC data of these fractions (Table 1). These results suggest that these fractions could contain high molecular weight molecules since it has been found that polyphenol type molecules with elevated molecular weight, such as procyanidins (polymers derived from flavan-3-ol monomer), exhibit a prominent antioxidant capacity [8,22,23]. The increased antioxidant capacity observed in the highly retained fractions on Sephadex LH-20 can be explained by two factors: the intrinsic activity of polyphenols that are present in the sample, or their abundance in avocado extracts. In many cases, a polyphenol can inherently be very bioactive, however it can be present in such a small quantity that it does not contribute to the antioxidant capacity observed in the sample. In this study, to avoid the concentration influence during measurement of antioxidant capacity, the same concentration and sample volume were used in all of the cases.

Based on the above-mentioned results, the fractions with the highest antioxidant activities (PSE, PPE, F2S, F3S, F2P, and F3P) were chosen to perform their chemical characterization by ultra-performance liquid chromatography–electrospray ionization–mass spectrometry (UPLC-ESI-MS/MS). 

### 2.2. Polyphenol Characterization by UPLC-ESI-MS/MS

Figure 2 shows the UPLC profiles of the PSE and the most retained seed fraction on Sephadex (F3S). Through mass spectrometry analyses (UPLC-ESI-MS/MS), different types of phenolic compounds, such as organic acids, phenolic acids, flavonoids and different condensed tannins (procyanidin dimers and trimers), were detected and identified as responsible for the in vitro antioxidant capacity previously measured in avocado peel and seeds.

High-performance liquid chromatography (HPLC), particularly reversed-phase HPLC, allowed a rapid and effective separation of polyphenols except for higher polymerized oligomers. Proanthocyanidins, flavan-3-ol monomers, dimers, trimers and their isomers were clearly separated by reversed-phase HPLC. Furthermore, highly polymerized oligomers appeared as broad unresolved peaks on the chromatogram because of the enormous variety of isomers and oligomers with different degrees of polymerization (Figure 2B).

The presence of polyphenolic compounds was verified based on the following parameters: (i) the characteristic absorbance (λ_max_), (ii) the measurement of the exact mass of the pseudomolecular ion [M − H]^−^ (which was compared to a database of standard polyphenols from previous studies, finding an observed mass error <10 ppm with respect to the theoretical mass), and (iii) the information previously published in the literature (Table 2).

Two organic acids, quinic and citric acids, were identified in PSE and PPE, as compounds **1** and **2**, respectively. These compounds have been previously identified in avocado seeds and peels [17,24]. 

Some flavonoids were identified. The flavan-3-ols, catechin and epicatechin (**9** and **15**), were found in phenolic-rich extracts as well as, and fractions F2S and F2P. The mass spectra of these compounds showed a pseudomolecular ion at *m*/*z* 289 [M − H]^−^, and the typical fragment ion at *m*/*z* 245 that correspond to the descarboxylation of the molecule [M – 44 − H]^−^. Konsinska et al. [15] have reported the presence of these isomers in Hass, Lamby, and Rugoro varieties. The flavone apigenin (**29**) and the flavonol kaempferol (**30**) were detected in PSE and PPE. In addition, the glycosylated chalcone phloridzin (florentin glucoside, **26**) was here reported for the first time in avocado (PSE, PPE, and F2S). The flavonol quercetin (**28**) was also found in PSE and PPE, and some of its glycosylated derivatives, quercetin diglucoside (**21**), quercetin 3-*O*-arabinosyl-glucoside (**22**), quercetin 3-*O*-glucoside (**23**), quercetin-3-*O*-rutinoside (**24**), quercetin-3-*O*-arabinoside (**25**), and quercetin 3-*O*-rhamnoside (**27**) were tentatively identified by mass spectrometry, mainly in the F2P fraction from avocado peels. The phenolic compounds quercetin 3-*O*-rhamnoside and quercetin-3-*O*-rutinoside were here detected for the first time in peel and seed avocado respectively. In all of their mass spectra, the characteristic ion fragments at *m*/*z* 301, 271, 243, 199, and 107 of aglycon quercetin were detected. For example, the ion fragment at *m*/*z* 301 in the quercetin-3-*O*-arabinosyl-glucoside (**22**) mass spectrum corresponds to the loss of one hexose and one pentose moieties [M – 162 – 132 − H]^−^, respectively; in the quercetin-3-*O*-rutinoside (**24**) to the loss of one hexose and one rhamnose moeites [M – 162 – 146 − H]^−^, respectively; in the quercetin-3-*O*-arabinoside (**25**), to the loss of a pentose moiety [M – 132 − H]^−^; in the quercetin-3-*O*-rhamnoside (**27**) mass spectrum to the loss of 146 u, characteristic of rhamnose moiety; and in the mass spectrum of compound **21** to the loss of two hexoses [M-162-162-H]^−^. The other ion fragments could be explained as follows: *m*/*z* 271 [301 − CH_2_O]^−^ as a result of a formaldehyde loss, *m*/*z* 243 due to the loss of carbon monoxide [271 − CO]^−^, *m*/*z* 199 because of the loss of carbon dioxide [243 − CO_2_]^−^, and *m*/*z* 107 corresponding to the A and B rings loss from quercetin. Kosinska et al. [15] have reported the presence of quercetin-3,4′-diglucoside and compounds **22** and **24** in peels of Hass variety, and compounds **23** and **25** in peels of Shepard avocado.

Phenolic acids were mainly found in the PSE and PPE extracts, thus suggesting that they remained in F1S and F1P fractions, which were not analyzed by UPLC-MS. The 5-O-caffeoylquinic acid (**13**, isomer of chlorogenic acid) mass spectrum showed an ion fragment at *m*/*z* 191 that corresponds to the loss of caffeic acid moiety [M – 162 − H]^−^, which is characteristic of the 5-OH binding [25]. The compound **13** has been already identified in peels and seeds of Hass and Shepard varieties [20], and in the pulp of Rugoro variety [21]. Similarly, caffeic acid (**14**), syringic acid (**7**), *p*-coumaric acid (**17**), ferulic acid (**18**), and sinapic acid (**19**) were here identified, and previously reported in avocado seeds and peels [15,17].

The fractionation of PSE and PPE extracts by Sephadex LH-20 allowed to obtain the compounds with high molecular weights, such as condensed flavonoids (proantocyanidins), in fractions F2 and F3. In the seeds, four procyanidin dimer isomeric compounds, type B (**5**, **8, 11,** and **20**) were detected, with pseudomolecular ions at *m*/*z* 577 [M − H]^−^, together the fragment ions at *m*/*z* 425, 407, 289, and 245. In each case, the fragment at *m*/*z* 425 [M-152-H]^−^ corresponds to a retro-Diels–Alder (rDA) fission of the heterocyclic rings of the procyanidins dimers; the fragment ion at *m*/*z* 407 is generated by a Diels–Alder fractionation in the C ring, with a subsequent elimination of a water molecule; the fragment at *m*/*z* 289 [M – 288 − H]^−^ is originated as a consequence of a methyl-quinone break of the interflavan bond, with the loss of an epicatechin molecule [26,27] and the subsequent elimination of CO_2_, generates the fragment ion at *m*/*z* 245 [15,28,29]. In the peel only two type B dimers were detected (compounds **5** and **11**). Due to the lack of standards, it was not able to establish the precise type of dimer (B_1_–B_4_). 

The type B dimeric procyanidins display C_4_–C_8_ or C_4_–C_6_ bonds, while those of type A show additional ether bonds between C_2_–*O*–C_5_ or C_2_–*O*–C_7_ [30,31]. Therefore, type A dimeric procyanidins (such as compound **3**, *m*/*z* 575) have a lower number of hydrogen atoms in their structure with respect to type B ones. Dimeric procyanidins have already been identified in seeds and/or peels of Hass variety [6,15,24,27,28]. Kosinka et al. [15] reported the presence of two type B procyanidin dimers (*m*/*z* 577) and another type A (*m*/*z* 575) in peels of the Hass variety.

Procyanidin trimers type A and type B (**4**, **6**, **10** and **12**) were also detected, specifically in F3 fractions of avocado seeds and peels. The isomeric compounds **4, 6** and **12** showed a pseudomolecular ion at *m*/*z* 865 [M-H]^−^ and ion fragments at *m*/*z* 577 [M – 288 − H]^−^, corresponding to the loss of a catechin unit. This ion was produced after a methyl-quinone cleavage of the [M − H]^−^ ion between the linkage of the upper and second units (Figure 3); at *m*/*z* 425 [M −288 – 152 − H]^−^ due to a retro-Diels–Alder (rDA) fission of the heterocyclic ring of the procyanidin dimers; and at *m*/*z* 289 corresponding to a catechin monomer, thus allowing the identification of these compounds as isomers of (epi)catechin–(epi)catechin–(epi)catechin. In contrast to the type B procianidin, methyl-quinone fission of type A procianidin requires breaking two bonds. The compound **10** is a type A trimeric procianidin (Figure 4) and exhibited a pseudomolecular ion at *m*/*z* 863 [M − H]^−^ that is 2 u less than the corresponding to type B procianidin trimer. The fragment ion at *m*/*z* 575 [M – 288 − H]^−^ was produced after methyl-quinone cleavage of [M − H]^−^ between the linkage of the central and terminal unit (F and G rings, Figure 4). Thus, the terminal unit was identified as (epi)catechin and the upper unit (ion at *m*/*z* 575) as a type A dimer. The fragment ion at *m*/*z* 287 [M – 2 × 288 − H]^−^ was produced by cleavage of the ion at *m*/*z* 575 between the linkage of the first and second units of the ion at *m*/*z* 575. The compounds **4**, **6**, **10** and **12** were detected in seeds and compounds **10** and **12** were only detected in peels. Kosinska et al. [15] identified type A and B procyanidin trimers in Hass and Shepard varieties. These authors found only two type A trimers (*m*/*z* 863) in seeds, whereas in this study four procyanidin trimers were detected in avocado seed; three type B procyanidins trimers (*m*/*z* 865) and one type A (*m*/*z* 863).

To summarize, the chemical analysis of this study revealed that seed extract of Nariño cultivar avocado contained flavonols, catechins, hydroxycinnamic acids, quercetin glycosides and procyanidins (Figure 5). For the first time, phloridzin was detected in avocado seed and peels, as well as five procyanidins dimers (compounds **3**, **5**, **8**, **11**, and **20**) and four procyanidins trimers (compounds **4**, **6**, **10**, and **12**) in seeds. In the peels, two procyanidins dimers (compounds **5** and **11**) and two procyanidins trimers (compounds **10** and **12**) were detected. The presence of type A procyanidins (PCs) in avocado peel could provide additional health benefits to these avocado by-products, since this type of compounds have proven to be more resistant to microbial catabolism [32]. The results of the antioxidant capacity showed that F2 and F3 fractions of avocado seeds and peels exhibited the highest antiradical capacity. These fractions contain the procyanidin dimers and trimers, which antioxidant capacity has been associated to polymerization degree [33,34,35]. Many of the benefits attributed to procyanidins are related to their chemical characteristics. It has been reported that procyanidins can modulate antioxidant enzymatic activities [8], exhibit chemoprotective properties against cancer [9,36], are anti-*Helicobacter pylori* agents [35], improve lipid metabolism [8], and prevent urinary tract infections [36], among other bioactivities.

## 3. Materials and Methods 

### 3.1. Chemicals and Reagents

Folin–Ciocalteu reagent, all solvents (methanol, acetone and acetonitrile), and standards (gallic acid, Trolox and ABTS) were purchased from Sigma-Aldrich (St. Louis, MO, USA). Quinic acid, citric acid, procyanidin dimer B_1_ and B_2_, catechin, caffeic acid, vanillin, epicatechin, syringic acid, *p*-coumaric acid, ferulic acid, sinapic acid, phloridzin, quercetin, kaempferol and apigenin, were obtained from Sigma-Aldrich (St. Louis, MO, USA) and Fluka (St. Louis, MO, USA). 

### 3.2. Plant Material

*Persea americana* Mill. avocado samples were collected in the municipality of Sandoná (Nariño, Colombia) at 1848 m above sea level, and a fruit specimen was stored in the University of Nariño Herbarium (Code No. 13691). A total of 14 fruits of uniform size (8.2 × 7.1 cm), bright green coloration, and in an optimal state for consumption (pH 6.12, 6.07 °Brix) were collected through a simple random sampling. The fruits were carefully washed with deionized water, weighted, and then their peels and seeds were manually separated. After homogenization process, 600 g of peels and 600 g of seeds were processed and the samples kept at 4 °C until analyzed. 

### 3.3. Preparation of Avocado Byproduct Extracts

The crude extracts were obtained following the method described by Kosinska et al. [15]. The fruit parts were cut into small pieces (1 cm^2^) and 200 g were extracted by maceration with 250 mL 80% methanol for 24 h at 15 °C. After filtration (Whatman filter No. 1), the organic solvent was evaporated under vaccum and the resulting extract was redissolved in deionized water and lyophilized (Labconco, MO, USA). The lyophilized samples, named as crude seed extract (CSE, 11.06 g) and crude peel extract (CPE, 6.56 g), were maintained at 4 °C until their analyses. The extraction process was carried out in triplicate and data was presented as mean ± standard deviation. The same process was carried out using 70% aqueous acetone. For selection of the best solvent, data was analyzed using Statgraphics Centurion 16.1.15 program using the total phenolic content (CFT) and the Trolox equivalent antioxidant capacity (TEAC) as variable responses.

### 3.4. Polyphenolic-Rich Fractions of Avocado Byproducts

Initially, the crude extracts (CSE and CPE) obtained with 70% aqueous acetone were separately adsorbed on an open column (24 × 2 cm i.d.) loaded with 150 g a nonionic polymeric absorbent (Amberlite XAD-7; Sigma-Aldrich, St. Louis, MO, USA). Separated portions (2 g, 6 g in total) were washed with 500 mL deionized water until the complete elution of sugars (^o^Brix). The phenolic-rich fraction in each case was eluted with 300 mL of methanol-acetic acid (19:1, *v*/*v*); based on the procedure reported by Degenhardt et al. [37]. The solvent was removed under vaccum and the residue redissolved in water and lyophilized. Polyphenol-rich extracts PSE (phenolic-rich seed extract, 3.15 g), and PPE (phenolic-rich peel extract, 3.93 g) were thus obtained.

For the subsequent fractionation of PSE and PPE, the procedure described by Yang et al. [38] was followed. The sample (0.25 g of each polyphenol-rich extract) was first dissolved in water/methanol (70:30, *v*/*v*), and this solution was subjected to size exclusion chromatography by using a Sephadex LH-20-filled column (26 × 2 cm). The fraction with the least retained compounds was eluted with 400 mL of water/methanol (70:30, *v*/*v*), collecting 32 fractions of 10 mL each that showed the same UV absorption profile. These fractions were combined and named F1S (90 mg) for PSE and F1P (48 mg) for PPE. The compounds with intermediate retention times were eluted with 450 mL of methanol/water (60:40, *v*/*v*), allowing the collection of 37 fractions, 10 mL each, which were combined and named F2S (149.3 mg) and F2P (161.3 mg), respectively. Finally, the most retained fractions (17 fractions of 10 mL each) were eluted with 250 mL of acetone/water (60:40, *v*/*v*), combined and named F3S (470.3 mg) and F3P (471 mg), respectively. All of these procedures were repeated three times.

### 3.5. Antioxidant Capacity Evaluation

The samples were dissolved with 70% methanol to obtain a final assay mixture concentration of 50 mg/mL, all extracts were evaluated at this concentration. Values are presented as mean ± standard deviation of three determinations.

#### 3.5.1. Total Polyphenolic Content (TPC)

The total polyphenolic content of the *Persea Americana* Mill. extracts, was assessed by the Folin–Ciocalteau method [39]. A 100 μL aliquot from each sample was separately mixed with 900 μL of Folin–Ciocalteau (Sigma-Aldrich, St. Louis MO, USA) reagent, and this solution was kept at 18 °C for 5 min. Then, 750 μL sodium bicarbonate was added, and this solution was stirred for 30 s and maintained at 18 °C for 90 min. Finally, the absorbance at λ 765 nm was measured by using a Merck Spectroquant^®^ Pharo 300 spectrophotometer (Darmstadt, Germany) and the results were expressed as milligram of gallic acid equivalents (GAE) per 100 g of dried extract.

#### 3.5.2. ABTS Radical Scavenging Capacity

The antioxidant capacity was in vitro measured based on the ability to capture the ABTS radical [40], which is generated by the ABTS (7 mM) oxidation with potassium persulfate (2.45 mM) in water. The ABTS solution was diluted in phosphate buffer (SBF) at pH 7.4 and then kept for 16 h at 18 °C, in darkness until reaching an absorbance equal to 0.70 ± 0.02 at λ 734 nm. Then, 3 mL of ABTS diluted solution was added to 30 μL of each sample and mixed for 1 min. After 6 min, the absorbance was measured at λ 734 nm in a spectrophotometer. Final results were obtained by interpolation of the absorbance into the Trolox calibration curve (0.5–3.0 mM) and were expressed as mmol of Trolox equivalents/grams of dried extract.

#### 3.5.3. DPPH Radical Scavenging Capacity

The procedure established by Sánchez-Moreno et al. [41] was used to estimate the effect of each extract on a DPPH radical solution. An aliquot of each sample (0.1 mL) was added to 3.9 mL of DPPH methanolic solution (0.025 g/L) prepared on the same day. The absorbance at λ 515 nm was measured in a spectrophotometer at different time intervals until the solution reached a silver color (steady state). The DPPH concentration in the reaction medium was calculated by using the following calibration curve: A_515_ = 12.25 [DPPH]_T_ − 0.001624, where [DPPH]_T_ was expressed as g/L (*r* = 0.9985). The remnant DPPH percentage (%DPPH_REM_) was calculated as follows: %DPPH_REM_ = [DPPH]_T_/[DPPH]_T = 0_. The %DPPH_REM_ was plotted versus the antioxidant concentration and used to obtain the amount of antioxidant necessary to decrease the DPPH initial concentration until 50% (EC_50_).

### 3.6. UPLC-ESI-MS/MS Analysis of Phenolic Compounds

The polyphenolic compounds present in the most active fractions were analyzed by high-performance liquid chromatography (UPLC Dionex Ultimate 3000 RS, Waltham, MA, USA) coupled to a Hybrid Quadrupole-Orbitrap Mass Spectrometer (QExactive) with HESI (Heated Electrospray Ionization source in negative mode. Prior to direct injection, the samples were filtered through a 0.45 μm nylon filter (Sigma Aldrich, St. Louis, MO, USA). Separation of the compounds was achieved with a Xbridge BEH C_18_ column (100 mm × 2.1 mm, 2.5 μm) through a binary solvent system consisting of a phase A (H_2_O at 0.1% in trifluoroacetic acid) and phase B (methanol at 0.1% in trifluoroacetic acid) [24]. The following gradient was applied: 5% B (1 min), 5–100% B (10min), 100% B (2 min), 5% B (3min); the flow rate was 0.5 mL/min. The column temperature was 40 °C. The injection volume for all selected fractions was 20 µL. The instrument was operated in negative ion mode using a scan range from *m*/*z* 50 to 1000. Nitrogen was used as the dry gas at a flow rate of 4.5 mL/min, the temperature was set at 250 °C and the collision energy (HCD cell) was operated at 30 kv. Detection was based on calculated exact mass and on retention time of target compounds.

### 3.7. Statistical Analysis

All statistical analyses were carried out using the Statgraphics Centurion 16.1.15 software (Virginia, VA, USA). The univariant variance analysis (ANOVA) using a general linear model program was applied to establish whether the mean values of the sample data differed significantly from each other. The mean values from each set of samples (*n* = 3) were compared using the Fisher’s least significant difference (LSD) procedure.

## 4. Conclusions

The fractionation method described here is potentially useful for the characterization of polymerized polyphenols and the isolation of oligomeric constituents. The bioguided separation used (size exclusion chromatography) was effective to isolate fractions with high molecular weight polyphenolic compounds, such as dimers and trimers of proantocyanins. These fractions exhibited the highest values of antioxidant capacity under ABTS and DPPH bioassays, and correlated with a higher polyphenolic compound content (TPC).

The fractionation of the phenols allowed for grouping them according to their size and the ability to scavenge free radicals. Statistical differences were found within the phenolic content and capacity antioxidant of seed and peel; it was found that the fractions of higher average molecular weight have significant antiradical capacity in comparison with the previously reported data. The results of this work showed that *Persea americana* Mill. byproducts (seeds and peels) are a rich source of bioactive compounds exhibiting antioxidant capacity. It was established that the Nariño cultivar seed contains more phenolic compounds and is more antioxidant compared to other varieties of avocado already studied in literature. The presence of procyanidins type A, with potential health benefits, was found in peel and seeds. These results showed the potential use of avocado extracts for pharmaceutical and food industries due to the huge content of bioactive compounds.

## Figures and Tables

**Figure 1 molecules-24-03209-f001:**
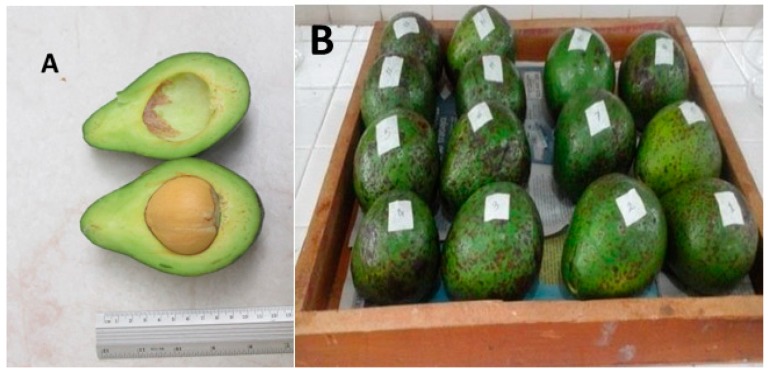
Avocado (*Persea americana*) (**A**) pulp and (**B**) fruit from Sandoná (Nariño, Colombia).

**Figure 2 molecules-24-03209-f002:**
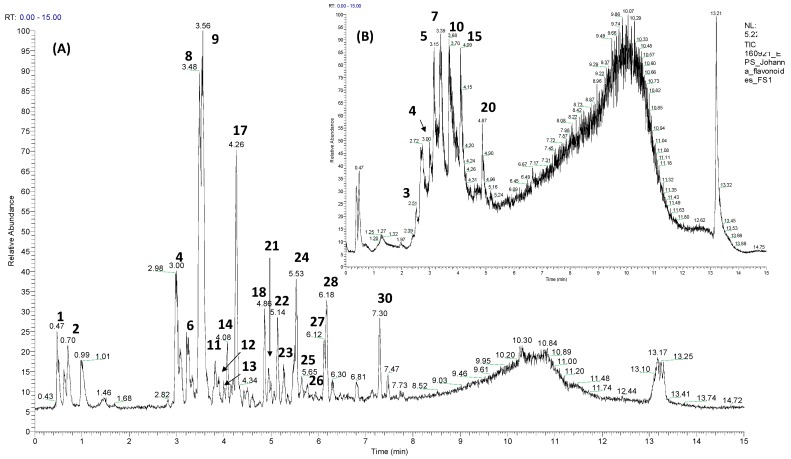
UPLC-TIC (ultra HPLC total ion current) chromatograms of (**A**) the phenolic-rich seed extract (PSE), and (**B**) F3S fraction. Peak numbers refer to those indicated in Table 2.

**Figure 3 molecules-24-03209-f003:**
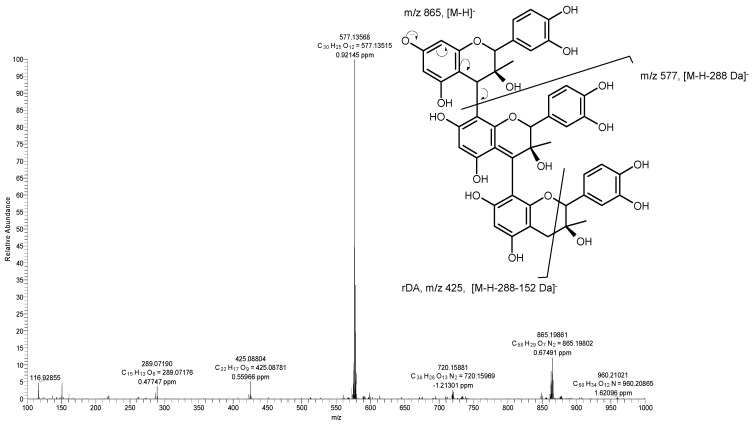
Mass spectrum and molecular structure of procyanidin trimer B-isomer 1 (compound **4)**.

**Figure 4 molecules-24-03209-f004:**
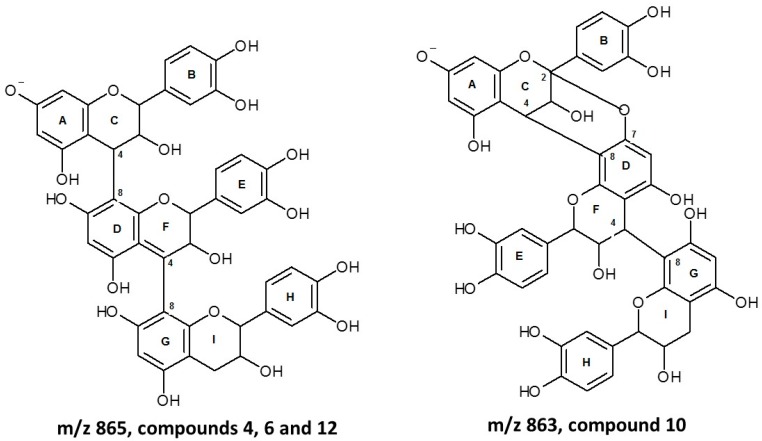
Type A and type B procyanidin trimers identified in *Persea americana* Mill. Nariño cultivar.

**Figure 5 molecules-24-03209-f005:**
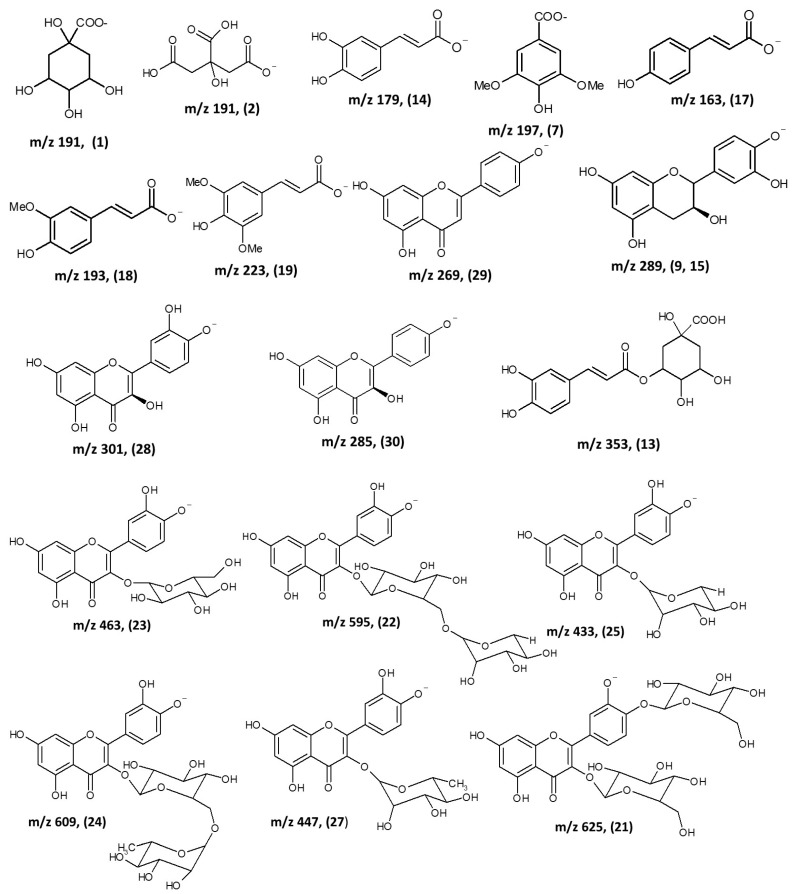
Organic acids, phenolic acids, flavonoids and procyanidin dimers identified in *Persea americana* Mill. Nariño cultivar.

**Table 1 molecules-24-03209-t001:** Comparison of antioxidant capacity results of seeds and peels avocado (*Persea americana* Mill. Nariño cultivar) extracts and fractions.

Sample	TPC (mg Gallic Acid Equivalents (GAE)/g Dried Extract) *	TEAC (mmol Trolox/g Dried Extract) *	DPPH (IC_50_ g Antioxidant/ kg DPPH) *
*Avocado seed*			
CSE	328.8 ± 13.5 ^a^	3.2 ± 0.2 ^a^	320.1 ± 12.1 ^a^
PSE	1303.0 ± 67.7 ^b^	18.4 ± 0.9 ^b^	90.1 ± 4.5 ^b^
F1S	588.4 ± 35.5 ^c^	7.6 ± 0.5 ^c^	191.0 ± 9.0 ^c^
F2S	926.9 ± 30.7 ^d^	12.6 ± 0.7 ^d^	104.3 ± 4.8 ^d^
F3S	752.2 ± 33.2 ^e^	12.1 ± 0.7 ^d^	86.3 ± 6.1 ^b^
*Avocado peel*			
CPE	527.8 ± 22.1 ^f^	5.7 ± 0.3 ^e^	138.2 ± 3.6 ^e^
PPE	1058.0 ± 59.7 ^g^	12.1 ± 0.7 ^d^	82.5 ± 4.1 ^f^
F1P	363.8 ± 18.9 ^h^	4.8 ± 0.3 ^e^	373.5 ± 15.4 ^g^
F2P	1050.9 ± 59.8 ^g^	13.7 ± 0.7 ^f^	84.7 ± 4.9 ^f^
F3P	963.6 ± 38.7 ^i^	11.9 ± 0.6 ^d^	79.7 ± 5.3 ^f^

* Values are expressed as mean ± SD (*n* = 3). TPC = total polyphenolic contents; TEAC = Trolox equivalent antioxidant capacity. Values in the same column followed by different letters are significantly different by ANOVA test (*p* < 0.05). CSE = crude seed extract; CPE = crude peel extract; PSE = phenolic-rich seed extract; PPE = phenolic-rich peel extract; F1S, F2S, and F3S = fractions obtained from PSE; F1P, F2P, and F3P = fractions obtained from PPE.

**Table 2 molecules-24-03209-t002:** Characterization of Phenolic Compounds in Fractions of *Persea americana* Mill. Nariño cultivar Byproducts (seeds and peels). For peak assignation see Figure 2.

Peak No.	Rt (min)	Tentative Identification	UV–Vis ʎ_max_ (nm)	*m*/*z* Measured [M − H]^−^	*m*/*z* Theoretical [M − H]^−^	Molecular Formula	MS/MS Fragment Ions (*m*/*z*)	PSE	F2S	F3S	PPE	F2P	F3P
1	0.47	Quinic acid *	265	191.0563	191.0561	C_7_H_12_O_6_	127.0400	x	-	-	x	-	-
2	0.70	Citric acid *	230	191.0199	191.0197	C_6_H_8_O_7_	111.0088	x	-	-	x	-	-
3	2.51	Procyanidin dimer A	279	575.1199	575.1971	C_30_H_24_O_12_	289.0720	-	-	x	-	-	-
4	3.00	Procyanidin trimer B-isomer 1	279	865.1986	865.7645	C_45_H_38_O_18_	577.1356, 425.0880, 289.0719, 245.0820	-	-	x	-	-	
5	3.15	Procyanidin dimer B1 *	280	577.1354	577.1352	C_30_H_26_O_12_	425.0122, 407.0774, 289.0722, 245.0822	x	x	x	x	x	-
6	3.22	Procyanidin trimer B-isomer 2	277	865.2003	865.7645	C_45_H_38_O_18_	289.0722, 125.0244	-	-	-	-	-	x
7	3.39	Syringic acid *	-	197.0455	197.0456	C_9_H_10_0_5_	123.0088	-	-	-	x	-	-
8	3.48	Procyanidin dimer B-isomer 2	280	577.1352	577.1352	C_30_H_26_O_12_	407.0769, 289.0722, 245.0822	x	x	-	-	-	-
9	3.56	Catechin *	277	289.0717	289.0718	C_15_H_14_O_6_	245.0819	x	x	-	x	x	-
10	3.68	Procyanidin trimer A	279	863.1844	863.7486	C_45_H_36_O_18_	575.1350, 287.0722	-	-	x	-	-	x
11	3.79	Procyanidin dimer B2 *	279	577.1353	577.5123	C_30_H_26_O_12_	425.0778, 289.0718, 245.0721	-	-	x	-	-	x
12	3.80	Procyanidin trimer B-isomer 3	278	865.2002	865.7645	C_45_H_38_O_18_	577.1354, 425.0879, 289.0718 245.0721	-	-	x	-	-	-
13	3.91	5-*O*-caffeoyl-quinic acid	325	353.0878	353.0878	C_16_H_18_0_9_	191.0211	x	-	-	x	-	-
14	4.08	Caffeic acid *	320	179.0349	179.0350	C_9_H_8_O_4_	107.0502, 135.0451	x	-	-	x	-	-
15	4.09	Epicatechin *	276	289.0720	289.0718	C_15_H_14_O_6_	245.0819	x	x	-	x	x	-
16	4.18	Vanillin *	269	151.0400	151.0401	C_8_H_8_O_3_	108.0217, 136.0166	x	-	-	x	-	-
17	4.26	*p*-Coumaric acid *	308	163.0400	163.0401	C_9_H_8_O_3_	119.0502	x	-	-	x	-	-
18	4.86	Ferulic acid *	320	193.0506	193.0506	C_10_H_10_O_4_	149.0608, 134.0373	x	-	-	x	-	-
19	4.72	Sinapic acid *	325	223.0611	223.0612	C_11_H_12_O_5_	-	x	-	-	x	-	-
20	4.87	Procyanidin dimer B-isomer 4	277	577.1357	577.5123	C_30_H_26_O_12_	245.0819, 289.0718	-	-	x	-	-	-
21	4.96	Quercetin diglucoside	279, 235	625.1415	625.5090	C_27_H_30_O_17_	301.0279, 271.0253, 243.0301, 199.0405, 107.0139	-	-	-	-	x	-
22	5.21	Quercetin 3-*O*-arabinosyl-glucoside	280, 235	595.1308	595.4830	C_26_H_28_O_16_	301.0279, 271.0252, 243.0299, 199.0404, 107.0141	-	-	-	-	x	-
23	5.38	Quercetin-3-*O*-glucoside	355	463.0888	463.0882	C_21_H_20_O_12_	301.0278, 271.0250, 243.0300	x	-	-	x	x	-
24	5.53	Quercetin-3-*O*-rutinoside (rutin)	320	609.1469	609.1461	C_27_H_30_O_16_	301.0280, 271.0252, 199.0406, 107.0141	x	-	-	x	x	-
25	5.62	Quercetin 3-*O*-arabinoside	279	433.0777	433.3424	C_20_H_18_O_11_	301.0278, 271.0187, 243.0311	-	-	-	-	x	-
26	5.86	Phloridzin *	283	435.1296	435.1297	C_21_H_24_O_10_	273.0769, 167.0350	x	x	-	x	-	-
27	6.12	Quercetin 3-*O*-rhamnoside	279, 234	447.0934	447.3690	C_21_H_20_O_11_	301.0280, 271.0252, 255.0302, 227.0352, 151.0038, 107.0140	-	-	-	-	x	-
28	6.18	Quercetin *	370	301.0356	301.0354	C_15_H_10_O_7_	151.0037, 121.0295	x	-	-	x	-	-
29	7.03	Apigenin *	278	269.0456	269.0456	C_15_H_10_O_5_	117.0346	x	-	-	x	-	-
30	7.30	Kaempferol *	278	285.0403	285.0406	C_15_H_10_0_6_	185.0608	x	-	-	x	-	-

* Rt and MS compared to the corresponding standards. PSE = phenolic-rich seed extract; PPE = phenolic-rich peel extract; F2S, and F3S = fractions obtained from PSE; F2P and F3P = fractions obtained from PPE.

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
