# Peer review of "Analysis of Phenolic Composition of Byproducts (Seeds and Peels) of Avocado (Persea americana Mill.) Cultivated in Colombia"

_molecules, 2019, doi:10.3390/molecules24173209_

Reviewer 1 Report

Despite of the quality and extensive research work showed in this manuscript, authors need to justify the novelty of this research. 

See for instance:

https://doi.org/10.1021/jf300090p

https://doi.org/10.3923/rjmp.2017.55.61

DOI: 10.1016/j.indcrop.2016.03.005.

DOI: 10.1016/j.lwt.2016.06.049.

DOI: 10.1016/j.foodres.2017.11.082.

DOI: 10.1016/j.foodchem.2017.12.011.

DOI: 10.1021/acs.jafc.5b03099.

https://doi.org/10.1021/jf1048832

Other suggestions:

Line 28. Improve the introduction section.

¿How relevant is the avocado byproducts production? ¿How much is generated in Colombia or worldwide? 

¿How about avocado byproduct extracts and in vivo assays? Recently, in vitro assays have been very questioned as compared to the real effect of bioactive compounds on humans (¿Are they really bioavailable?). 

Line 94. Need to correct spelling

Line 95. Figure 1 would be improved if authors add scales

Line 129. Table 1. Move text to footprint

Line 139. Check grammar

Line 298. Table 2. Add meaning of PSE, F2S, F3S, PPE, F2P, F3P in footprint.

Author Response

Reviewer 1:

Open Review

English language and style

( ) Extensive editing of English language and style required 
( ) Moderate English changes required 
( ) English language and style are fine/minor spell check required 
(x) I don't feel qualified to judge about the English language and style 

Yes

Can be improved

Must be improved

Not applicable

Does the introduction provide sufficient   background and include all relevant references?

(x)

( )

( )

( )

Is the research design appropriate?

(x)

( )

( )

( )

Are the methods adequately described?

(x)

( )

( )

( )

Are the results clearly presented?

(x)

( )

( )

( )

Are the conclusions supported by the results?

(x)

( )

( )

( )

Comments and Suggestions for Authors

The article entitled Bioguided Analysis of Antioxidant Compounds from Byproducts (Seeds and Peels) of Avocado (Persea americana Mill.) Cultivated in Colombia is an interesting one. The authors presents in a clear way their findings and their findings are well correlated with the literature. In my opinion the article do not present lacks in term of quality. I believe that the article should be accepted after minor corrections.

The abbreviaton from the tables must be explained in the table title or below the table

R// Abbreviations used in tables were explained in the corresponding footprints (lines 177-180 for Table 1, and lines 400-401 for Table 2).

Reviewer 2 Report

Avocado fruits are most of all a valuable food product possessing health-promoting values, a rich source of highly-valued oil containing appreciable levels of unsaturated fatty acids, phytosterols and vitamin E. This oil occupies also an important position in cosmetology, including aromatherapy, as the vehicle for essential oils. The oil is accumulated in the mesocarp (pulp).

The authors highlight the possibility to use other fruit parts, avocado pulp byproducts, i.e. seeds and peels as a rich source of antioxidants.

It is a very interesting and valuable experimental work showing experience of the authors and their professional approach as regards the methods used, data presentation, discussion of the results, etc. Although studies on the contents of antioxidants in avocado seeds and peels were published earlier by other research teams with the use of other cultivars (??) of avocado, the authors of the present paper were successful in identification of new compounds in the Nariño cultivar. Moreover, by fractionation of the extracts the authors proved that the highest antioxidant activity was exhibited by high molecular weight fractions. The authors should be given credit for the scale of research as they demonstrated the presence of 30 polyphenolic compounds in the studied extracts.

Recommendations as to particular chapters of the manuscript and expected supplements and improvements are listed below.

Abstract:

-          -line 15 – note! There is a factual error there since the Folin-Ciocalteau method is used to determine the total content of polyphenols but not antioxidant activity!!!

-          lines 19/20 – „catechins” have been omitted

-          lines 21/22 „and” is repeated twice

Key words:

-          UPLC-MS should be replaced by UPLC-ESI/MS/MS

Introduction:

-          an information about the significance of avocado fruits in cosmetology and aromatherapy should be supplemented.

-          line 77 – „catequin” should be replaced by „catechin”

Results & Discussion:

-          The authors are affiliated to chemical not biological or botanical institutions and interchangeably use the terms: subspecies, varieties, cultivars, however, the meaning of these biological terms is different! The authors should carefully check the whole manuscript to assure the correct use of these terms. It is very important!

see line 149 – cultivars, while in l. 152 – varieties, again l. 156 – cultivar, see l. 485 – cultivar, but line 486 – varieties (from a botanical and genetic viewpoint these are very serious errors!!!)

-          line 94 – a word is missing 15.5% (?) 9.3%

-          line 160 – „It’s” should be replaced by „It was”

-          line 312 – „hidroxycinnamic” should be replaced by „hydroxycinnamic”

-          line 321/322 – a predicate is missing in the sentence

-          line 339/340 – „HPLC water” is a lab jargon, it should be replaced by a professional term

Figures and Tables

Table 2:

-          The title lacks the name: „Nariño” cultivar, which is included in Fig. 4 and Fig. 5 captions.

-          compound no. 13 – the correct separate of the name the compound is: 5-O-caffeoyl / quinic acid

-          compound no. 24 – „rutin” should be replaced by „rutoside”

-          There is RŁ in the title and TR under the table, it should be unified !

Fig. 4:

-          The full stop is lacking: Mill, should be replaced by Mill. (it is an abbreviation of the name of the taxonomist who described the studied species, so the lack of the full stop is a serious error!)

Conclusions:

-          line 483, again no full stop after Mill (see comment to Fig. 4)

-          the information that the obtained results indicate also the possibility to use the studied extracts of seeds and peels in the cosmetic industry, can be supplemented

Language assessment

The paper is generally well written, some minor mistakes are listed below:

l. 15     their antioxidant properties were measured

l. 17/18 By UPLC-17 ESI/MS/MS, The polyphenolic-rich extracts and their fractions were analyzed by UPLC-17 ESI/MS/MS, finding

l. 29     coma is unnecessary

l. 30     to Mexico

l. 34/35             the climatic requirements and zones of origin of which have been

l. 53     to a lesser extent

l. 131   are significantly different

l. 133   Similar findings were found – please rephrase

l. 161   had the highest

l. 202   oligomers were appeared as broad unresolved peaks

l. 221   reported by for the first time

l. 244   extracts, such thus suggesting

etc.

Please check the manuscript for other possible mistakes.

Author Response

Reviewer 2:

Open Review

English language and style

( ) Extensive editing of English language and style required 
( ) Moderate English changes required 
(x) English language and style are fine/minor spell check required 
( ) I don't feel qualified to judge about the English language and style 

Yes

Can be improved

Must be improved

Not applicable

Does the   introduction provide sufficient background and include all relevant   references?

( )

( )

(x)

( )

Is the research   design appropriate?

(x)

( )

( )

( )

Are the methods   adequately described?

( )

( )

(x)

( )

Are the results   clearly presented?

( )

(x)

( )

( )

Are the conclusions   supported by the results?

( )

(x)

( )

( )

Comments and Suggestions for Authors

Despite of the quality and extensive research work showed in this manuscript, authors need to justify the novelty of this research. 

R// This justification was improved in lines 101-106. Almost all of the references suggested by this referee, were already cited in the manuscript, see numbers below.

See for instance:

https://doi.org/10.1021/jf300090p Kosinska et al [20]

https://doi.org/10.3923/rjmp.2017.55.61 (this reference was included in discussion, see lines 222-225 and reference [21])

DOI: 10.1016/j.indcrop.2016.03.005 (this reference was not included because the aim is focused on the extraction; other papers of the same authors related to the measurement of antioxidant activity in avocado seed had been included in the manuscript [13 and 17])

DOI: 10.1016/j.lwt.2016.06.049 López-Cobo et al.[28]

DOI: 10.1016/j.foodres.2017.11.082. Figueroa et al. [24]

DOI: 10.1016/j.foodchem.2017.12.011. Figueroa et al. [25]

DOI: 10.1021/acs.jafc.5b03099. (This reference was not included because it was done in avocado fruit pulp)

https://doi.org/10.1021/jf1048832 Rodríguez-Carpena et al. [6]

Other suggestions:

Line 28. Improve the introduction section.

¿How relevant is the avocado byproducts production? ¿How much is generated in Colombia or worldwide? 

R// These data were included in lines 63-66.

¿How about avocado byproduct extracts and in vivo assays? Recently, in vitro assays have been very questioned as compared to the real effect of bioactive compounds on humans (¿Are they really bioavailable?). 

R// The introduction was modified according to the referee´s recommendation. See lines 94-97.

Line 94. Need to correct spelling

R// It was corrected in line 116.

Line 95. Figure 1 would be improved if authors add scales

R// It was done in line 117. The first photo in figure 1 was changed for another that includes the scale.

Line 129. Table 1. Move text to footprint

R// Ok, it was done, lines 177-180

 Line 139. Check grammar

R// It was done in line 182.

Line 298. Table 2. Add meaning of PSE, F2S, F3S, PPE, F2P, F3P in footprint.

R// It was done in lines 177-180 for Table 1, and lines 400-401 for Table 2.

Sincerely,

Nelson Hurtado

Associated Professor

Departamento de Química

Universidad de Nariño

Reviewer 3 Report

The article entitled Bioguided Analysis of Antioxidant Compounds from 2 Byproducts (Seeds and Peels) of Avocado (Persea 3 americana Mill.) Cultivated in Colombia is an interesting one. The authors presents in a clear way their findings and their findings are well correlated with the literature. In my opinion the article do not present lacks in term of quality. I believe that the article should be accepted after minor corrections.

The abbreviaton from the tables must be explained in the table title or below the table

Author Response

Reviewer 3:

Open Review

English language and style

( ) Extensive editing of English language and style required 
( ) Moderate English changes required 
(x) English language and style are fine/minor spell check required 
( ) I don't feel qualified to judge about the English language and style 

Yes

Can be improved

Must be improved

Not applicable

Does the   introduction provide sufficient background and include all relevant   references?

( )

(x)

( )

( )

Is the research   design appropriate?

(x)

( )

( )

( )

Are the methods   adequately described?

(x)

( )

( )

( )

Are the results   clearly presented?

(x)

( )

( )

( )

Are the conclusions   supported by the results?

(x)

( )

( )

( )

Comments and Suggestions for Authors

Avocado fruits are most of all a valuable food product possessing health-promoting values, a rich source of highly-valued oil containing appreciable levels of unsaturated fatty acids, phytosterols and vitamin E. This oil occupies also an important position in cosmetology, including aromatherapy, as the vehicle for essential oils. The oil is accumulated in the mesocarp (pulp).

The authors highlight the possibility to use other fruit parts, avocado pulp byproducts, i.e. seeds and peels as a rich source of antioxidants.

It is a very interesting and valuable experimental work showing experience of the authors and their professional approach as regards the methods used, data presentation, discussion of the results, etc. Although studies on the contents of antioxidants in avocado seeds and peels were published earlier by other research teams with the use of other cultivars (??) of avocado, the authors of the present paper were successful in identification of new compounds in the Nariño cultivar. Moreover, by fractionation of the extracts the authors proved that the highest antioxidant activity was exhibited by high molecular weight fractions. The authors should be given credit for the scale of research as they demonstrated the presence of 30 polyphenolic compounds in the studied extracts.

Recommendations as to particular chapters of the manuscript and expected supplements and improvements are listed below.

Abstract:

-line 15 – note! There is a factual error there since the Folin-Ciocalteau method is used to determine the total content of polyphenols but not antioxidant activity!!!

R// We agree with you. We corrected this mistake in lines 15 and 16.

-          lines 19/20 – „catechins” have been omitted

R// it was included in line 20

-          lines 21/22 „and” is repeated twice

R// It was corrected in line 22.

Key words:

-          UPLC-MS should be replaced by UPLC-ESI/MS/MS

R// It was corrected in line 27.

Introduction:

-          an information about the significance of avocado fruits in cosmetology and aromatherapy should be supplemented.

R// This was including in lines 72-73

-          line 77 – „catequin” should be replaced by „catechin”

R// It was corrected, line 90.

Results & Discussion:

-          The authors are affiliated to chemical not biological or botanical institutions and interchangeably use the terms: subspecies, varieties, cultivars, however, the meaning of these biological terms is different! The authors should carefully check the whole manuscript to assure the correct use of these terms. It is very important!

see line 149 – cultivars, while in l. 152 – varieties, again l. 156 – cultivar, see l. 485 – cultivar, but line 486 – varieties (from a botanical and genetic viewpoint these are very serious errors!!!)

R// Cultivars is OK in line 206. Ref. 19 Cultivars line 164.

The Nariño cultivar was corrected through all text (lines 114, 173, 213, 218, 396, 410, 413, 424, 603)

-          line 94 – a word is missing 15.5% (?) 9.3%

R// The word to was included in line 116.

-          line 160 – „It’s” should be replaced by „It was”

R// It was corrected in line 227.

-          line 312 – „hidroxycinnamic” should be replaced by „hydroxycinnamic”

R// It was corrected in line 413.

-          line 321/322 – a predicate is missing in the sentence

R// A predicate was included in line 427.

-          line 339/340 – „HPLC water” is a lab jargon, it should be replaced by a professional term

R// it was corrected in line 444.

Figures and Tables

Table 2:

-          The title lacks the name: „Nariño” cultivar, which is included in Fig. 4 and Fig. 5 captions.

R// Nariño cultivar was included in the title of table 2 (line 396)

-          compound no. 13 – the correct separate of the name the compound is: 5-O-caffeoyl / quinic acid

R// Name of compound 13 was corrected

-          compound no. 24 – „rutin” should be replaced by „rutoside”

R// We consider that rutin is a more known name. We did not change the name.

-          There is RŁ in the title and TR under the table, it should be unified !

R// Rt was unified.

Fig. 4:

-          The full stop is lacking: Mill, should be replaced by Mill. (it is an abbreviation of the name of the taxonomist who described the studied species, so the lack of the full stop is a serious error!)

R// It was corrected in the line 409.

Conclusions:

-          line 483, again no full stop after Mill (see comment to Fig. 4)

R// It was corrected in line 601.

-          the information that the obtained results indicate also the possibility to use the studied extracts of seeds and peels in the cosmetic industry, can be supplemented

R// We don´t have enough evidence to point out that. In cosmetic industry, avocado oils are commonly used, we obtained polar extracts.

Language assessment

The paper is generally well written, some minor mistakes are listed below:

l. 15     their antioxidant properties were measured

R// it was corrected in line 15

l. 17/18 By UPLC-17 ESI/MS/MS, The polyphenolic-rich extracts and their fractions were analyzed by UPLC-17 ESI/MS/MS, finding

R// It was corrected in line 19.

l. 29     coma is unnecessary

R// It was corrected in line 31.

l. 30     to Mexico

R// It was corrected line 32.

l. 34/35             the climatic requirements and zones of origin of which have been

R// It was corrected in line 37

l. 53     to a lesser extent

R// It was corrected in line 63

l. 131   are significantly different

R// It was corrected in line 177

l. 133   Similar findings were found – please rephrase

R// It was rephrased in line 182.

l. 161   had the highest

R// It was corrected in line 218.

l. 202   oligomers were appeared as broad unresolved peaks

R// It was corrected in line 274.

l. 221   reported by for the first time

R// It was corrected in line 295.

l. 244   extracts, such thus suggesting

R// It was corrected in line 318.

Please check the manuscript for other possible mistakes.

R// Yes, it was done.

Sincerely,

Nelson Hurtado

Associated Professor

Departamento de Química

Universidad de Nariño
